# Service availability and readiness of malaria surveillance information systems implementation at primary health centers in Indonesia

**Maria Holly Herawati[1], Besral[2], Dina Bisara Lolong[1], Noer Endah Pracoyo[1], Noor Edi Widya Sukoco[1], Hadi Supratikta[1,3], Meita Veruswati** 📷[4,5]*, **Al Asyary** 📷[6]

1 National Research and Innovation Agency, Jakarta, Indonesia, 2 Department of Biostatistics and Population Studies, Faculty of Public Health, Universitas Indonesia, Depok, Indonesia, 3 Study Program of Management Studies, Postgraduate School, Universitas Pamulang, South Tangerang, Banten, Indonesia, 4 Study Program of Public Health, Faculty of Health Sciences, Universitas Muhammadiyah Prof. Dr. HAMKA, Jakarta, Indonesia, 5 PhD in Business and Management, Postgraduate School, Management and Science University, Shah Alam, Selangor Darul Ehsan, Malaysia, 6 Department of Environmental Health, Faculty of Public Health, Universitas Indonesia, Depok, Indonesia

* meitaveruswati@uhamka.ac.id

**Data Availability Statement:** Data contain potentially identifying or sensitive patient

## Abstract

One of the most important indicators in malaria eradication is the malaria surveillance information system (SISMAL) for recording and reporting medical cases. This paper aims to describe the availability and readiness of SISMALs at primary health centers (PHCs) in Indonesia. A cross-sectional survey was implemented in seven provinces for this study. The data was analyzed using bivariate, multivariate, and linear regression. The availability of the information system was measured by assessing the presence of the electronic malaria surveillance information system (E-SISMAL) at the studied PHCs. The readiness was measured by averaging each component of the assessment. From 400 PHC samples, only 58.5% had available SISMALs, and their level of readiness was only 50.2%. Three components had very low levels of readiness: (1) the availability of personnel (40.9%), (2) SISMAL integration and storage (50.2%), and (3) the availability of data sources and indicators (56.8%). Remote and border (DTPK) areas had a 4% better readiness score than non-DTPK areas. Endemic areas were 1.4% better than elimination areas, while regions with low financial capacity were 3.78% better than regions with high financial capacity, with moderate capacity (2.91%). The availability rate of the SISMAL at PHCs is only 58.5%. Many PHCs still do not have SISMALs. The readiness of the SISMAL at these PHCs is significantly related to DTPK/remote area, high endemicity status, and low financial capacity. This study found that the implementation of SISMAL is more accessible to malaria surveillance for the remote area and regions with low financial capacity. Therefore, this effort will well-fit to address barrier to malaria surveillance in developing countries.

information, data are owned by a state organization. Data are available from the Health Policy and Development Board, Indonesian Ministry of Health (BKPK-Kemkes RI) Institutional Data Access / Ethics Committee (contact datin. bkpk@kemkes.go.id) for researchers who meet the criteria for access to confidential data.

**Funding:** This research was supported by the National Institute for Research and Development, Indone-sian Ministry of Health which has been integrated into Indonesian National Research and Inno-vation Agency (BRIN). No additional external funding was received for this study. The funders had no role in study design, data collection and analysis, decision to publish, or preparation of the manuscript.

**Competing interests:** The authors have declared that no competing interests exist.

## 1. Introduction

As a tropical country, Indonesia is endemic to tropical diseases such as malaria [1, 2]. The malaria control program has been in effect since 1914, during the Dutch colonial period [3]. Indonesia National Health Day was first held on November 12, 1959, when President Sukarno symbolically sprayed pesticide against malaria mosquitoes in the Kalasan village, a special region of Yogyakarta Province [4]. The 2019 malaria control report states that out of 514 districts/cities in Indonesia, 300 districts were declared eliminated (free of malaria) (58%), 160 were slightly endemic (31%), 31 were moderately endemic (6%), and 23 were highly endemic (5%) [5].

The malaria control program in Indonesia aims, by 2030, to achieve total malaria elimination (malaria free) in stages. There are three main steps that consisted of decrease Annual Parasite Incidence (API) and Positivity Rate (PR), and increase Annual Blood Examination Rate (ABER). The steps taken are to grant a malaria certification per region, starting from the district/city to the provincial, regional, and national levels. The term "regional certification" arises because the area of Indonesia is huge, and thus, the capacities of health services, population, and regional infrastructure are varied. According to these, it means that the main requirement for malaria free comprised of: (1) there is no indigenous case for three consecutive years, (2) PR less than 5%, and (3) API less than 1 of 100,000 rate. However, the imported case is not considering for these steps. There are five areas of malaria regionalization, covering the Java and Bali regions, Sumatra, the West Nusa Tenggara and Sulawesi regions, the Kalimantan and North Maluku regions, the NTT and Maluku regions, and the West Papua and Papua regions [5].

The malaria control program was developed to expand the public health effort. Good-quality health development requires system strengthening. According to the World Health Organization (WHO), an aspect of such strengthening is the information system [6]. This will increase the validity and completeness of malaria data reporting. According to Law No. 4 of 1984, malaria is an endemic infectious disease with the potential to be an epidemic. Therefore, epidemiological recording or surveillance reports must be supported on an ongoing basis [7, 8]. The legal basis for strengthening the information system of malaria data reporting is stated in the decree of the minister of health of the Republic of Indonesia (Number 293/2009) regarding the elimination of malaria in Indonesia. It was stated in the sixth strategy, i.e., to organize a surveillance system, with monitoring and evaluation, as well as a health information system for malaria. Moreover, the third pillar of the Global Technical Strategy for Malaria 2016–2030, the transformation of malaria surveillance, is a core indicator of the importance of strengthening such a system [9, 10].

The use of information systems in the malaria control program is essential in the capacity building of the program [9, 11]. Information systems guarantee the accuracy of data collection and analysis to support decision making. The information system governance approach is a strategic effort used as the basis for policies at various levels of health services and at all levels of government in response to the challenges of the times. In 2012, an electronic-based application of the malaria surveillance information system (SISMAL) was developed for the recording and reporting of malaria control. This application has been implemented in several districts. These districts, including Kepulauan Seribu (Jakarta), Bali, and Batam, had been previously set as the first of four-phase the elimination region goals in 2030, [12, 13]. Then in 2015, the electronic malaria surveillance information system (E-SISMAL) software application was implemented in health services. Each districts have to input/compile and delivery their report monthly by this system, The E-SISMAL software was established to improve validity and completeness in reporting malaria data [10]. One of the requirements for a district/city to obtain a malaria elimination certificate is a complete malaria case register, which includes a complete elimination area [11, 14, 15].

Since 2018, the second version of the E-SISMAL has been disseminated to all provinces. This version utilizes a web-based system to improve performance and data quality [10].

This study aims to assess and discuss the availability and readiness of the information system for malaria control programs in health-care centers, i.e., the SISMAL. The data was analyzed from National Health Facility Research (Rifaskes) in combination with data collected from a thematic study of health information systems in 2019 [16].

## 2. Materials and methods

A cross-sectional survey was implemented in this study. Data was collected from seven purposively selected provinces, with the specific consideration of certain factors, i.e., the regional representativeness of BPJS Kesehatan (the Health Care and Social Security Agency) and different territories of Indonesia. These provinces are Aceh, West Java, West Nusa Tenggara, West Kalimantan, Central Sulawesi, South Sulawesi, and Papua. From the 7 provinces, we randomly (with proportional-clustered random sampling) selected 103 districts/cities and 400 health centers. The collected data through quantitative survey of cross-sectional study was using by structured questionnaire that includes items that was had validity and reliability test. This instrument was tested in the different health centers (n = 30) that shows >0.3 in all item using item-total correlation (valid) as well as >0,6 of Cronbach's alpha value (reliable). The rational of readiness score was developed from the theory of organizational readiness for change [17]. We interviewed the head and staff of the Malaria program at each health-care center [16]. We then recruited a local interviewer who had been involved with the Rifaskes in 2019. Training sessions for the interviewer were held virtually and during face-to-face meetings. Direct supervision by the research team was carried out during data collection [16].

The data was analyzed using logistic predictive model to analyze the availability of the information system. The statistical analysis was started with bivariate (p-value <0.25) as the screening test for eligible predictors. These eligible predictors were carried on with multivariate by analysis of co-variance (ANCOVA) regression analysis. The availability of the information system was measured by assessing the presence or absence of the E-SISMAL at primary health centers (PHCs) by its significant value of <0.05; 95% CI. Data analysis was performed based on province, remote area (DTPK), regional financial capacity (<0.445 = lowest and low; 0.445–0.807 = moderate; ≥0.808 = high and highest [18]), and regional status according to the malaria program, and each area was classified as either eliminated or endemic (low, medium, high, or combined). The readiness of the SISMAL was measured by averaging each component of the assessment, which includes organization, availability of personnel, availability of infrastructure, data sources and indicators, monitoring and evaluation data and verification, data quality, presentation, utilization and feedback, data integration, and storage data.

The procedures followed were in accordance with the ethical standards of the committee on human experimentation (institutional and national) (Protocol Code: Leader of Centre of Research and Development for Human Resources and Health Services, National Institute for Health Research and Development, Indonesian Ministry of Health, No. HK.03.02/1/180/2019, and Ethical Approval LB.02.01/2/KE.186/2019) [16]. Informed consent was obtained from all subjects involved in the study.

## 3. Results

### 3.1. Availability of malaria surveillance information systems at primary health centers

Of all the 400 PHCs in 7 provinces, only 58.5% had malaria program information systems available. The three provinces with the lowest availability of SISMALs at PHCs were West Java (27.6%), West Kalimantan (50.0%), and Papua (54.5%). Meanwhile, the three provinces with

**Table 1. Availability of malaria surveillance information systems at primary health centers (n = 400).**

| Characteristics | Available | | Not Available | | Total |
|---|---|---|---|---|---|
| | N | % | N | % | |
| **Thematic Province** | | | | | |
| West Java | 34 | 27.6 | 89 | 72.4 | 123 |
| West Kalimantan | 14 | 50.0 | 14 | 50.0 | 28 |
| West Papua | 6 | 54.5 | 5 | 45.5 | 11 |
| Aceh | 34 | 64.2 | 19 | 35.8 | 53 |
| Central Sulawesi | 19 | 73.1 | 7 | 26.9 | 26 |
| South Sulawesi | 106 | 77.4 | 31 | 22.6 | 137 |
| West Nusa Tenggara | 21 | 95.5 | 1 | 4.5 | 22 |
| **Region** | | | | | |
| DTPK | 46 | 82.1 | 10 | 17.9 | 56 |
| Non-DTPK | 188 | 54.7 | 156 | 45.3 | 344 |
| **Regional Financial Capacity** | | | | | |
| Low | 125 | 62.2 | 76 | 37.8 | 201 |
| Moderate | 81 | 54.7 | 67 | 45.3 | 148 |
| High and very high | 28 | 54.9 | 23 | 45.1 | 51 |
| **Endemic Status** | | | | | |
| Elimination (low) | 159 | 54.6 | 132 | 45.4 | 291 |
| Endemic (medium and high) | 75 | 68.8 | 34 | 31.2 | 109 |
| Total | 234 | 58.5 | 166 | 41.5 | 400 |

the highest availability of SISMALs at PHCs were West Nusa Tenggara (95.5%), South Sulawesi (77.4%), and Central Sulawesi (73.1%).

The availability of the SISMAL differs greatly by regions. PHCs in DTPK areas had a higher level of information system availability than PHCs in non-DTPK areas. Of the 56 PHCs in DTPK areas, 82.1% had available SISMALs. Meanwhile, of the 344 PHCs in non-DTPK areas, only 54.7% had available SISMALs.

The availability of the SISMAL also varies considerably according to regional financial capacity. PHCs with low regional financial capacity had a higher level of information system availability than PHCs with moderate or high regional financial capacity. Of the 201 PHCs with low financial capacity, 62.2% had available SISMALs. Meanwhile, only 54% of the PHCs with moderate or high financial capacity had available SISMALs.

Lastly, the availability of the SISMAL differs significantly according to the endemicity status of the region. PHCs in endemic areas had a higher level of information system availability than PHCs in elimination areas. Of the 109 PHCs in endemic areas, 68.8% had available SISMALs. Meanwhile, of the 291 PHCs in elimination areas, only 54.6% had available SISMALs (Table 1).

## 3.2. Readiness of malaria surveillance information systems at primary health centers

SISMAL availability at the PHC does not guarantee readiness in the operational activities of processing and analyzing data as well as presenting and utilizing information related to malaria. For the 234 PHCs (58.5%) with available SISMALs, the level of readiness to run the system only reached 50.2% (117 PHCs). Of the eight components of the SISMAL readiness assessment, three components had very low levels of readiness on average: (1) availability of

SISMAL personnel (40.9% or 96 PHCs), (2) SISMAL integration and storage (50.2% or 117 PHCs), and (3) availability of data sources and indicators (56.8% or 133 PHCs).

Various problems related to SISMAL personnel at the PHCs included the unavailability of such personnel with the required educational background and the absence of regulations regarding their transfer. Only 22 (9.4%) PHCs of SISMAL personnel had educational backgrounds in the fields of statistics, epidemiology, computers, or medical records, and the availability of regulations regarding transfers was only 23 (9.8%) PHCs. Although 183 PHCs already had a permanent SISMAL staff rate (78.2%), only 155 (66.2%) PHCs had received formal training to manage SISMALs.

The main problem related to the unpreparedness of the data sources and indicators is that currently, their level of readiness is only 56.8% or 133 PHCs. The components that need the top priority for improvement are those that cannot describe the input, process, and output indicators (only 72 or 30.8% of the studied PHCs are able to describe them).

Various problems related to SISMAL integration and data storage are that the SISMAL does not use the National Identification Number (NIK) code or the International Classification Code (ICD) code (only 58 or 24.6% PHCs use NIKs/ICDs), that no bridging exists with other data sources (only 121 or 51.7% PHCs do data bridging), and that the document retention period is still less than five years (only 80 (34.3%) PHCs have kept their documents for five years or more). Meanwhile, 212 or 90.6% of PHCs archive their data electronically or in the cloud.

Organizational SISMAL readiness has reached 165 (70.4%) PHCs. The components that need improvement are organizational structure and job descriptions and functions (readiness of only 106 or 45.3% of PHCs). Only 143 or 61.1% of PHCs already have SISMAL organizational structures as well as job descriptions and functions. On the bright side, most PHCs already have special persons in charge of the SISMAL sector (194 or 82.9%).

Readiness in the fields of facilities and infrastructure is also low, currently only 146 (62.6%) PHCs. Of the five components of facilities and infrastructure, three require priority repairs: the availability of standard operational procedures (SOPs) (79 or 33.8% of PHCs), the availability of computers and local area networks (LANs) (123 or 52.6% of PHCs), and the availability of special funds for the SISMAL (144 or 61.5% of PHCs). The other two components, internet availability (181 or 77.4% of PHCs) and the availability of SISMAL recording standards (201 or 85.9% of PHCs), still have good standing.

Readiness in the fields of data monitoring, evaluation, and verification is very good at 218 (93.2%) PHCs. Of the 234 PHCs with SISMAL availability, almost 100% (98.7% or 231 PHCs) have involved external parties in monitoring and evaluating data, with 97.9% performing data verification activities. NIK, ICD, and CBG verification has reached 212 (90.6%) PHCs.

Readiness in the field of data quality is also very good at 221 (94.4%) PHCs. All the data quality components have readiness levels between 218 (93.2%) to 224 (95.8%) PHCs, which include routine, complete, and timely reporting as well as easy-to-retrieve reporting data.

Readiness in the fields of data presentation, utilization, and feedback has reached 211 (90.3%) PHCs. The component that needs improvement is report feedback (186 or 79.5% of PHCs). The other components still have good standing in terms of readiness, namely, the way to present data (229 or 97.9% of PHCs readiness), the type of media to disseminate information (225 or 96.2% of PHCs readiness), and the use of information by other agencies (210 or 89.6% of PHCs readiness) (Table 2).

**Table 2. Readiness of malaria surveillance information systems at primary health centers (n = 234).**

| Readiness of SISMAL in Puskesmas | n | % |
|---|---|---|
| **Organization** | **165** | **70.4** |
| • Basic policy of information system: Permenkes + other | 216 | 92.3 |
| • There is an organizational structure as well as job descriptions and functions | 106 | 45.3 |
| • There is an organizational structure of the information system | 143 | 61.1 |
| • There is a person in charge of the information system | 194 | 82.9 |
| **Human Resources** | **96** | **40.9** |
| • Information systems educational background (statistics/epidemiology/computer/medical records) | 22 | 9.4 |
| • There is a regulation on the transfer of information systems personnel | 23 | 9.8 |
| • There is formal training for the person in charge of the information system | 155 | 66.2 |
| • There is permanent staff for the information system | 183 | 78.2 |
| **Availability of Infrastructure** | **146** | **62.6** |
| • There are standard operating procedures for presentation and others | 79 | 33.8 |
| • There are available computers and LANs | 123 | 52.6 |
| • There is a special fund for the information system | 144 | 61.5 |
| • There is internet availability | 181 | 77.4 |
| • There is standard recording of the information system | 201 | 85.9 |
| **Data Sources and Indicators** | **133** | **56.8** |
| • The information system can describe IPOs and other indicators | 72 | 30.8 |
| • Data source: Health service + network | 194 | 82.9 |
| **Monitoring, Evaluation, and Verification** | **218** | **93.2** |
| • MONEV data of information system: Health official + BPJS (moved) | 231 | 98.7 |
| • There is verification of internal/external data | 229 | 97.9 |
| • There has been verification of NIKs, ICDs, CBGs, and others | 212 | 90.6 |
| • There is data verification by the head of the center/director | 200 | 85.5 |
| **Data Quality** | **221** | **94.4** |
| • There are routine data reports | 218 | 93.2 |
| • There is complete reporting | 220 | 94.0 |
| • There is timely reporting | 222 | 94.9 |
| • The reported data is accurate and easy to obtain | 224 | 95.8 |
| **Presentation, Utilization, and Feedback** | **211** | **90.3** |
| • How to present data (text, tables, graphs, maps) | 229 | 97.9 |
| • Information media type: Print + online | 225 | 96.2 |
| • There is report feedback | 186 | 79.5 |
| • There is the use of information by other institutions | 210 | 89.6 |
| **Integration and Data Storage** | **117** | **50.2** |
| • The information system already uses the NIK/ICD code | 58 | 24.6 |
| • Length of data stored: 5+ years | 80 | 34.3 |
| • Bridging with data: Health official + others | 121 | 51.7 |
| • Archiving: Electronic/cloud | 212 | 90.6 |
| **Total Readiness Score** | 117 | **50.2** |

## 3.3. Readiness of malaria surveillance information systems at primary health centers per province

Out of the seven studied provinces, only two had SISMAL readiness levels above 60%, namely, West Nusa Tenggara and Central Sulawesi. The other five provinces had SISMAL readiness levels of around 50% to 60%, namely, Aceh, West Java, West Kalimantan, South Sulawesi, and Papua.

**Table 3. Readiness of malaria surveillance information systems at primary health centers by province (n = 234).**

| Readiness | Aceh | West Java | West Nusa Tenggara | West Kalimantan | Central Sulawesi | South Sulawesi | Papua |
|---|---|---|---|---|---|---|---|
| | n = 34 | n = 34 | n = 21 | n = 14 | n = 19 | n = 106 | n = 6 |
| K1. Organization | 49.3 | 55.7 | 65.5 | 46.4 | 63.2 | 44.1 | 62.5 |
| K2. Human Resources | 44.9 | 42.9 | 41.7 | 32.1 | 51.3 | 37.3 | 50.0 |
| K3. Infrastructure | 71.8 | 72.4 | 75.2 | 57.1 | 66.3 | 54.3 | 53.3 |
| K4. Indicator | 51.0 | 34.3 | 33.3 | 47.6 | 40.4 | 34.9 | 22.2 |
| K5. Verification | 56.9 | 60.8 | 61.9 | 52.4 | 70.2 | 54.7 | 47.8 |
| K6. Quality | 94.1 | 90.3 | 94.0 | 92.9 | 84.2 | 97.9 | 80.0 |
| K7. Utilization | 58.8 | 64.2 | 73.8 | 42.9 | 63.2 | 54.0 | 48.3 |
| K8. Integration | 44.9 | 47.4 | 45.2 | 41.1 | 46.1 | 48.1 | 44.2 |
| **K. Total** | **58,9** | **53,2** | **61,3** | **51,6** | **60,6** | **53,2** | **49,8** |

At the national level, the following components are of particular concern: (1) availability of SIK personnel, (2) availability of data and indicators, and (3) integration of information systems. The foremost problem related to SIK personnel is the unavailability of permanent SIK personnel who are trained and who have educational backgrounds in the fields of statistics, epidemiology, computers, and/or medical records. Meanwhile, the main problem with data and indicators is that the indicators produced have not been able to describe the input, process, and output components. Problems related to information system integration are that the SIS-MAL does not use NIK and ICD codes, that no bridging exists with other data sources, the data is not filed electronically, and the document storage time is still less than five years.

Apart from the three main issues mentioned above, certain provinces still have problems with the lack of support for the SISMAL's organizational structure, especially in Aceh, West Kalimantan, and South Sulawesi. The provinces that still have difficulty with the use of data and information are West Kalimantan and Papua. Papua also has difficulty with two other areas, namely, infrastructure and verification related to data and information (Table 3).

### 3.4. Readiness of malaria surveillance information systems at primary health centers by endemicity, DTPK/remote area, and financial capability

The results of the data analysis showed a normal distribution of SISMAL readiness with a mean value of 55.2% and a median of 53.2% (range 49.8%–61.2%). The ANCOVA analysis showed that SISMAL readiness at PHCs was significantly related to DTPK area, endemicity status, and regional financial capacity (p-value < 0.001; Coef.B = 50.56). The SISMAL readiness scores of the districts in remote areas were 4% better than those of districts in non-remote areas (p-value < 0.001; Coef.B = 4.22). The scores of endemic areas were 1.4% better than those of non-endemic areas. Lastly, the scores of regions with low and moderate financial capacity were better than those of regions with higher regional financial capacity, i.e., 3.78% (p-value < 0.001; Coef.B = 3.78) and 2.91% (p-value < 0.001; Coef.B = 2.91), respectively (Table 4).

## 4. Discussion

Our study shows that the PHCs with the highest scores of information system readiness were in the West Nusa Tenggara (NTB) province (95.5%), which is committed to achieving malaria elimination by 2019 [19]. Hence, the implementation of a malaria surveillance and reporting program is a priority in the province. Nowadays, NTB province is one of Indonesia province that becomes success malaria elimination which had already achieved the target within their 4

**Table 4. Results of ANCOVA multivariate analysis on the readiness of malaria surveillance information systems at primary health centers.**

| Parameter | Coef.B | Sig | Partial Eta squared |
|---|---|---|---|
| Intercept | 50.56 | <0.001 | 0.979 |
| DTPK | | | |
| DTPK vs Non-DTPK | 4.22 | <0.001 | 0.306 |
| Regional financial capacity | | | |
| Low | 3.78 | <0.001 | 0.192 |
| Moderate | 2.91 | <0.001 | 0.114 |
| High (ref) | | | |
| Endemicity | | | |
| Endemic vs Elimination | 1.4 | <0.001 | 0.073 |

[1] R-square = 0.493

of 10 regions. The governor of NTB province has also signed a memorandum of agreement with the Indonesian Ministry of Health in order to achieve the target in their regions in 2025.

The provinces with fairly good availability (60%–80%) were South Sulawesi (77.4%), Central Sulawesi (73.1%), and Aceh (64.2%). This result was supported by a number of studies [20–22]. A performance report in 2017 on South Sulawesi reported that malaria cases have been found in majority of districts/cities, i.e., twenty districts/cities with no mortality cases [20]. The malaria incidence rate per 1,000 population was targeted at less than 1 [23]. Until the fourth quarter of the reported year, the number of cases was 0.14 per 1,000 population [20]. In 2019, some regions in Central Sulawesi had high endemicity, but other districts achieved malaria elimination, such as Palu, Sigi, Buol, and Toli-toli [21]. Contrarily, in Aceh, only one area was stated as eliminated, while the others had low to high endemicity [22].

The availability and readiness of the SISMAL were the determinants of the success of the program. We must consider some management elements of the SISMAL as well as the program itself. The support of the central and local governments as well as city districts in controlling malaria with a variety of activities will determine the success of reducing malaria morbidity. Some activities carried out according to the malaria elimination manual are as follows [7]: (1) conducting mass blood surveys (MBSs), (2) conducting epidemiological investigations on each positive case of malaria, (3), carrying out malaria treatments according to standards, (4) conducting malaria vector control surveys [24], (5) conducting migration surveillance, (6) mapping receptive areas, (7) conducting cross-checks on the results of laboratory examinations, and (8) carrying out efficient recording and reporting. These points are supported by recommendations from Nuraini et al. They stated that the training and practice of E-SISMAL use is to improve the officers' abilities through socialization and that the Lahat Health Office establishes partnerships with telecommunications companies to ensure the smooth running of the network. Developing specific guidelines for the E-SISMAL and a mobile phone–based SISMAL to speed up the malaria reporting process in the Lahat district are also advisable [25]. As the one of the highest-ranking populations that utilized the mobile phone as well as social media in the world, the mobile phone-based SISMAL will be easily embraced [26]. Previous evidences of mobile health application were massively initiated and connecting the system all over the Indonesia regions [27, 28]. Moreover, its effective and efficient use will also being encourage and to well-adopted in other similar settings countries [29–31].

The Annual Malaria Research in Indonesia (AMRI) seminar to exchange insights on the future of malaria surveillance, diagnosis, and treatment in the country (held on August 19,

2020) declared the use of online-based applications and digital-based approaches in the malaria program as a form of innovation. The Province Health Office (PHO) of East Nusa Tenggara presented the use of an electronic and online malaria readiness self-assessment tool as relevant to malaria elimination efforts. Using this tool, health workers could track the progress of malaria intervention activities from the PHC up to the province level, with the aim to achieve a malaria-free district [5].

The role of the local government is important as well [24]. In Central Sulawesi Province, the government enacts policies for malaria elimination, as stated in the Governor's Decree No. 20/2012 [21] (Decree of the Minister of Health of the Republic of Indonesia No. 293/MENKES/SK/IV/2009, dated April 28, 2009, regarding the elimination of malaria in Indonesia, and the Minister of Home Affairs Circular No. 443.41/465/SJ of 2010 regarding the implementation of the malaria elimination program in Indonesia) [8, 9, 15]. The requirements to obtain a malaria elimination certificate are as follows: (1) no cases of indigenous (local) transmission for three consecutive years, (2) an annual parasite incident (API) rate of <1‰ for three consecutive years, and (3) a slide positive rate (SPR) of <5% for three consecutive years [8, 9, 15].

The readiness of SISMALs in PHCs in major areas was below the national average (below 70.2%). Meanwhile, some areas achieved readiness rates approaching 60%, including West Nusa Tenggara, Central Sulawesi, and Aceh. The problems observed were human resources, data storage components and integration, and SISMAL indicators.

The issues related to human resources were education and human resource resistance. These findings were in accordance with the results of Darman Zayadan [8, 32–38]. However, the issues related to data integration and storage components were caused by the suboptimal use of the civil ID registration number, which worsened interoperability. These findings were in accordance with policy studies on the governance of health information system resources in the implementation of PMK No. 97 of 2015 [32]. One study shows that interoperability not only helps improve patient safety and quality of care but also reduces the time and resources spent on data format conversion between hospitals. Other studies focus on the importance of an integrated SISMAL [33, 39–43]. Even the existence of an information system equipped with GIS is highly recommended for various infectious diseases such as malaria [40, 44, 45].

The problems with regard to data sources and indicators were the indicators of the input and output processes of the program, as supported by previous studies [22, 30]. In one paper, this indicator is termed a "data warehouse" [33, 46]. The vivax working group meeting routine report states that several general data indicators are needed in controlling malaria [45]. We attempted to address this problem and found that the data and information related to environmental factors such as rainfall rate, vector data, etc. were excluded and could not be reported as input data.

Remote area and regions with low financial capacity were more accessible in terms of readiness and availability. Endemic areas had more SISMAL availability than eliminated areas. Malaria could occur in many DTPK areas that lack financial capacity. Thus, the SISMAL is needed for the implementation of the program.

## 5. Conclusions

The SISMAL is available at only 58.5% of PHCs. Of this percentage, only 56.0% are ready to run it well. SISMAL readiness at the PHCs was significantly related to DTPK area, endemicity status, and regional financial capacity. DTPK areas, endemic areas, and regions with low or medium financial capacity have better levels of SISMAL readiness than non-DTPK areas, areas of elimination, and regions with high financial capacity [41]. Some studies have demonstrated

that SISMAL feasibility could significantly improve access to low-cost, valuable, and safe diagnoses [47]. Traditional epidemiological surveillance in other studies and the approach presented in this paper are complementary, but a formal validation framework for case identification algorithms is necessary.

## Acknowledgments

Our gratitude to all field technical person in charge as well as field enumerators that were contributing in data collection. We also thank to all provincial and regional government including health offices in the respective area of research.

## Author Contributions

**Conceptualization:** Maria Holly Herawati, Hadi Supratikta.

**Data curation:** Maria Holly Herawati, Meita Veruswati, Al Asyary.

**Formal analysis:** Besral.

**Funding acquisition:** Maria Holly Herawati.

**Investigation:** Dina Bisara Lolong, Noer Endah Pracoyo, Noor Edi Widya Sukoco, Meita Veruswati.

**Methodology:** Maria Holly Herawati, Besral, Dina Bisara Lolong, Hadi Supratikta.

**Project administration:** Maria Holly Herawati.

**Resources:** Dina Bisara Lolong, Hadi Supratikta.

**Software:** Besral.

**Supervision:** Maria Holly Herawati, Dina Bisara Lolong, Noer Endah Pracoyo, Noor Edi Widya Sukoco, Meita Veruswati, Al Asyary.

**Validation:** Besral, Noer Endah Pracoyo, Noor Edi Widya Sukoco, Al Asyary.

**Visualization:** Besral.

**Writing – original draft:** Maria Holly Herawati, Meita Veruswati.

**Writing – review & editing:** Besral, Al Asyary.

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
