## [Decision Letter · Decision Letter 0]

1 May 2022

PONE-D-22-07674Service Availability and Readiness of Malaria Surveillance Information Systems Implementation at Primary Health Centers in IndonesiaPLOS ONE

Dear Dr. Veruswati,

Thank you for submitting your manuscript to PLOS ONE. After careful consideration, we felt that your manuscript requires substantial revision, following which it can possibly be reconsidered, thus governing the decision of a “major revision”. As requested by the reviewers, the authors need to address several concerns, particularly related to the data analysis and methods (insufficient methodology for readers). Por example, the reviewer suggests that perhaps a logistic predictive model to analyze the availability of the information system according to several predictors may be preferred. Finally, considering  the study  falls into very local aspects, the author should clarify about  its interpretation and generalization. For your guidance, a copy of the reviewers' comments was included below.

We look forward to receiving your revised manuscript.

Kind regards,

Luzia Helena Carvalho, Ph.D.

Academic Editor

PLOS ONE

Journal Requirements:

a) Did participants provide their written or verbal informed consent to participate in this study?

"This research was funded by NIHRD Ministry of Health Republic of Indonesia."

We note that you have provided funding information. However, funding information should not appear in the Funding section or other areas of your manuscript. We will only publish funding information present in the Funding Statement section of the online submission form. 

"This research was funded by NIHRD Ministry of Health Republic of Indonesia."

7. Your ethics statement should only appear in the Methods section of your manuscript. If your ethics statement is written in any section besides the Methods, please delete it from any other section.

Reviewers' comments:

Reviewer's Responses to Questions

**Comments to the Author**

1. Is the manuscript technically sound, and do the data support the conclusions?

Reviewer #1: Partly

Reviewer #2: Yes

2. Has the statistical analysis been performed appropriately and rigorously? 

Reviewer #1: No

Reviewer #2: No

3. Have the authors made all data underlying the findings in their manuscript fully available?

Reviewer #1: Yes

Reviewer #2: Yes

4. Is the manuscript presented in an intelligible fashion and written in standard English?

Reviewer #1: Yes

Reviewer #2: Yes

5. Review Comments to the Author

Reviewer #1: PONE-D-22-07674

This is a brief description of the implementation and usage of an electronic information system (SISMAL) for malaria surveillance activities in Indonesia.

Increasing the capacity to implement digital health has been prioritized in the 2030 Agenda for Sustainable Development, as it could play a major role in accelerating progress towards achieving universal health coverage (https://apps.who.int/gb/ebwha/pdf_files/WHA71/A71_20-en.pdf). This experience could be of interest for lower-middle income countries (LMICs) interested in programmatic information systems and to understand potential limitations for implementation. However, the work falls into very local aspects that can complicate its interpretation and generalization.

Please find below some comments by sections

INTRODUCTION

It is unclear to me if the electronic malaria surveillance information system (E-SISMAL) is linked to a global information system of the country or is dedicated (i.e: malaria only).

METHODS

Interviews

Are the items of the interview considered arbitrarily? have been used in previous studies? Any rationale for the “Readiness score”?

Analysis

Perhaps a logistic predictive model to analyze the availability of the information system (yes/no) according to several predictors may be preferred. What was the rationale to use a linear regression analysis here? - The multiplicity of models declared here “The data was analyzed using bivariate, multivariate, and linear regression analysis” may require further detail.

DISCUSSION

The “mobile phone–based SISMAL” solution to encourage case reporting should necessarily consider a comment on the mobile phone utilization in the country. Moreover, I would have expected a comment on other experiences being carried out in similar countries. This restrict the interest to the local.

Reviewer #2: The paper discusses the service availability and readiness of PHCs in malaria surveillance in Indonesia. This is an important reporting for the malaria control community.

All my comments are included within the pdf document provided, in the comments section. Please access this pdf and respond conclusively to all the comments.

In summary, my main concerns are as follows:

1. Study design not stated in the methods section

2. Please add a concluding statement in the abstract. What does these results mean? Why is this study important?

3. Clearly explain in the methods section, the sampling techniques used

4. The statistical analysis techniques used in the study needs to be clearly rewritten and reported

6. PLOS authors have the option to publish the peer review history of their article (what does this mean?). If published, this will include your full peer review and any attached files.

Reviewer #1: No

Reviewer #2: **Yes: **Collins Okoyo

---

## [Author Response · Author response to Decision Letter 0]

14 Nov 2022

>>Dear Dr. Luzia Helena Carvalho, Ph.D. - Academic Editor of PLOS ONE,

Thank you very much for your interest in our research’s manuscript. Our gratitude also to the Reviewers that have been made substantial and constructive reviews and suggestions to our manuscript. 

We are very well agreeing that the methodology in our paper is still less to present in our previous form of it. In this section, we have now presented the clear-defined methodology including data analysis and how we were treating the confounding/bias with our statistical analysis. Additionally, we have also had explain the possible external validity/generalization of our study in the current form of our revised manuscript, so we hope that it would be sufficient to justify in LMICs setting.

Please find attached our rebuttal point-by-point revision according to the journal requirements as well as the Reviewers comments.

Best wishes,

Meita Veruswati – Corresponding author

---

## [Decision Letter · Decision Letter 1]

2 Feb 2023

PONE-D-22-07674R1Service Availability and Readiness of Malaria Surveillance Information Systems Implementation at Primary Health Centers in IndonesiaPLOS ONE

Dear Dr. Veruswati,

Thank you for submitting your manuscript for review to PLoS ONE. After careful consideration, we feel that your manuscript will likely be suitable for publication if the authors revise it to address critical points raised now by the reviewer.  According to reviewer, there are some specific areas where further improvements would be of substantial benefit to the readers.A copy of the reviewers’ comments was included for your information.  

We look forward to receiving your revised manuscript.

Kind regards,

Luzia H Carvalho, Ph.D.

Academic Editor

PLOS ONE

Journal Requirements:

Reviewers' comments:

Reviewer's Responses to Questions

**Comments to the Author**

1. If the authors have adequately addressed your comments raised in a previous round of review and you feel that this manuscript is now acceptable for publication, you may indicate that here to bypass the “Comments to the Author” section, enter your conflict of interest statement in the “Confidential to Editor” section, and submit your "Accept" recommendation.

Reviewer #3: (No Response)

Reviewer #4: All comments have been addressed

2. Is the manuscript technically sound, and do the data support the conclusions?

Reviewer #3: Yes

Reviewer #4: Yes

3. Has the statistical analysis been performed appropriately and rigorously? 

Reviewer #3: Yes

Reviewer #4: Yes

4. Have the authors made all data underlying the findings in their manuscript fully available?

Reviewer #3: Yes

Reviewer #4: Yes

5. Is the manuscript presented in an intelligible fashion and written in standard English?

Reviewer #3: Yes

Reviewer #4: Yes

6. Review Comments to the Author

Reviewer #3: The manuscript presented by Meita Veruswati et al. reports on Service Availability and Readiness of Malaria Surveillance Information Systems Implementation at Primary Health Centers in Indonesia. The authors present a descriptive assessment of the availability and readiness of the electronic malaria surveillance information system (E-SISMAL).

In general, the manuscript is well written, interesting, and very useful to the malaria community. I have only few issues which need addressing/clarification.

Minor issues

1. The author should briefly describe in method, what malaria data is being collected by E-SISMAL system and the frequency of collection, is it daily, weekly or monthly? Does it collecting routine malaria data, case-based surveillance system (CBS) or non-routine? Is it the only system for malaria surveillance in Indonesia? If yes, how does malaria data being collected in the areas where E-SISMAL/SISMAL is not implemented?

This is probably beyond the scope of this manuscript, but at least to understand the process of malaria surveillance in Indonesia.

2. In line 25: I like the way the authors present the result by strata, but the authors should clarify how these strata (low, high, and moderate) financial capacity is defined. The definition can be included here or in the method.

3. Line 76; Author mentioned application implemented in several districts, please indicate the number of districts in which SISMAL was implemented. And how many districts were targeted. To understand the coverage for E-SISMAL implementation at that time (2012).

4. In Line 270; I advise author to include this reference from WHO

"WHO Malaria elimination framework 2017" https://apps.who.int/iris/bitstream/handle/10665/254761/9789241511988-eng.pdf

5. In discussion section: The author explained very well the result of SISMAL, however I feel that the discussion could be strengthened a bit i.e., To describe how the NMCP /government can improve the availability of SISMAL in across all PHC. example the SISMAL now is implemented as standalone/web app which operate within the specific facility. There is need for a government to find a best approach to role out the system to all health facilities and have standardize and centralized system for malaria surveillance across the country.

Reviewer #4: In this manuscript, Herawati et al. present the results of a descriptive study of a Malaria Epidemiological Information System (SISMAL) in Indonesia. Basically, the authors describe the proportions of information´s availability and readiness in 400 health units randomly chosen among those that provide primary health care in 7 provinces spread across the country.

I revised a version of the manuscript in which I found a tracked changes revision performed by reviewer 1, along with all response from the authors were showed to all his comments. All the reviewer´s comments were addressed by the authors.

Although it is an essentially descriptive study, the authors carried out a comparative analysis to assess the association between the proportions of availability and readiness of the information system (SISMAL) with some characteristics related to the provinces where the health units are located, such as location (remote and border área), regional financial capacity, and regional status according to the malaria program (under elimination or

The manuscript is well written, with correct and clear English. The study methodology is adequate to respond to the study objectives. The data are well analyzed and the conclusion of the study is adequate to the observed results. The only point that I consider necessary to clarify in the Abstract is the authors' statement that "the readiness of SISMAL in these UBS is significantly related to the DTPK area, endemicity status and financial capacity”. In this sentence, it is important to clarify which categories of these variables were, in fact, associated with SISMAL readiness. That is, what were the categories of variables that were associated? Central or remote location? Endemic or elimination area? High or low financial capacity?

7. PLOS authors have the option to publish the peer review history of their article (what does this mean?). If published, this will include your full peer review and any attached files.

Reviewer #3: **Yes: **Joseph J. Joseph

Reviewer #4: No

---

## [Author Response · Author response to Decision Letter 1]

1 Mar 2023

Reviewer #3: 

The manuscript presented by Meita Veruswati et al. reports on Service Availability and Readiness of Malaria Surveillance Information Systems Implementation at Primary Health Centers in Indonesia. The authors present a descriptive assessment of the availability and readiness of the electronic malaria surveillance information system (E-SISMAL).

In general, the manuscript is well written, interesting, and very useful to the malaria community. I have only few issues which need addressing/clarification.

>>Dear Reviewer #3: Dr. Joseph J. Joseph,

Thank you very much for your appreciation to our manuscript. Our gratitude also for your substantial and constructive reviews and suggestions to our manuscript. 

Please find below our rebuttal point-by-point revision according to your-minor revisions’ comments below.

Minor issues

1. The author should briefly describe in method, what malaria data is being collected by E-SISMAL system and the frequency of collection, is it daily, weekly or monthly? Does it collecting routine malaria data, case-based surveillance system (CBS) or non-routine? Is it the only system for malaria surveillance in Indonesia? If yes, how does malaria data being collected in the areas where E-SISMAL/SISMAL is not implemented?

This is probably beyond the scope of this manuscript, but at least to understand the process of malaria surveillance in Indonesia.

>> Thank you for your comment. We have already provided these information in the introduction section particularly in the paragraph 4, in order to understand the process of malaria surveillance in Indonesia. And, yes, we agree with you that add information about its frequency collection (i.e. monthly reported to E-SISMAL) (line 78), with routine CBS. Currently, E-SISMAL/SISMAL is only implementing in Indonesia, however, we have already discussed about other malaria information system with CBS in other countries such as in low-resource setting countries. Thank you.

2. In line 25: I like the way the authors present the result by strata, but the authors should clarify how these strata (low, high, and moderate) financial capacity is defined. The definition can be included here or in the method.

>> Thank you for your suggestion. We have now clarified these strata of district financial capacity index (IKFD) that set by Indonesian Ministry of Finance in the method section (line 114-115).

3. Line 76; Author mentioned application implemented in several districts, please indicate the number of districts in which SISMAL was implemented. And how many districts were targeted. To understand the coverage for E-SISMAL implementation at that time (2012).

>> Thank you. We have now added the regions that implemented as the first of four-phase of 2030 elimination program in Indonesia.

4. In Line 270; I advise author to include this reference from WHO

"WHO Malaria elimination framework 2017" https://apps.who.int/iris/bitstream/handle/10665/254761/9789241511988-eng.pdf

>> Done! Thank you.

5. In discussion section: The author explained very well the result of SISMAL, however I feel that the discussion could be strengthened a bit i.e., To describe how the NMCP /government can improve the availability of SISMAL in across all PHC. example the SISMAL now is implemented as standalone/web app which operate within the specific facility. There is need for a government to find a best approach to role out the system to all health facilities and have standardize and centralized system for malaria surveillance across the country.

>> Thank you. The essential effort of the government to establish SISMAL in all health facilities has been elaborate in the discussioin section particularly started in line 309. We acknowledge the need of potential improvement for this system including the utilization of GIS (line 330) as well as mobile phone-based (line 292). And yes, we agree with you about standardize the system through data source/input parameters and indicators (line 332) that we have identified as the problem of this surveillance information system in varies and huge of Indonesia’s setting.

Reviewer #4: 

In this manuscript, Herawati et al. present the results of a descriptive study of a Malaria Epidemiological Information System (SISMAL) in Indonesia. Basically, the authors describe the proportions of information´s availability and readiness in 400 health units randomly chosen among those that provide primary health care in 7 provinces spread across the country.

I revised a version of the manuscript in which I found a tracked changes revision performed by reviewer 1, along with all response from the authors were showed to all his comments. All the reviewer´s comments were addressed by the authors.

Although it is an essentially descriptive study, the authors carried out a comparative analysis to assess the association between the proportions of availability and readiness of the information system (SISMAL) with some characteristics related to the provinces where the health units are located, such as location (remote and border área), regional financial capacity, and regional status according to the malaria program (under elimination or

The manuscript is well written, with correct and clear English. The study methodology is adequate to respond to the study objectives. The data are well analyzed and the conclusion of the study is adequate to the observed results. 

>>Dear Reviewer #4,

>>Thank you very much for your appreciation to our manuscript. 

The only point that I consider necessary to clarify in the Abstract is the authors' statement that "the readiness of SISMAL in these UBS is significantly related to the DTPK area, endemicity status and financial capacity”. In this sentence, it is important to clarify which categories of these variables were, in fact, associated with SISMAL readiness. That is, what were the categories of variables that were associated? Central or remote location? Endemic or elimination area? High or low financial capacity?

>> Thank you for your comments. We have now provide its classification that is associated with the readiness of SISMAL into: “the readiness of SISMAL in these UBS is significantly related to the DTPK/remote area, high endemicity status and low financial capacity”. These classification are defined in method sections.

---

## [Decision Letter · Decision Letter 2]

27 Mar 2023

Service Availability and Readiness of Malaria Surveillance Information Systems Implementation at Primary Health Centers in Indonesia

PONE-D-22-07674R2

Dear Dr. Veruswati,

We’re pleased to inform you that your manuscript has been judged scientifically suitable for publication and will be formally accepted for publication once it meets all outstanding technical requirements.

Kind regards,

Luzia H Carvalho, Ph.D.

Academic Editor

PLOS ONE

Additional Editor Comments (optional):

Reviewers' comments:

Reviewer's Responses to Questions

**Comments to the Author**

1. If the authors have adequately addressed your comments raised in a previous round of review and you feel that this manuscript is now acceptable for publication, you may indicate that here to bypass the “Comments to the Author” section, enter your conflict of interest statement in the “Confidential to Editor” section, and submit your "Accept" recommendation.

Reviewer #3: All comments have been addressed

2. Is the manuscript technically sound, and do the data support the conclusions?

Reviewer #3: Yes

3. Has the statistical analysis been performed appropriately and rigorously? 

Reviewer #3: Yes

4. Have the authors made all data underlying the findings in their manuscript fully available?

Reviewer #3: Yes

5. Is the manuscript presented in an intelligible fashion and written in standard English?

Reviewer #3: Yes

6. Review Comments to the Author

Reviewer #3: Overall, the manuscript is well-crafted, captivating, and highly valuable to the community involved in malaria research. The author(s) have effectively addressed all the concerns I raised

7. PLOS authors have the option to publish the peer review history of their article (what does this mean?). If published, this will include your full peer review and any attached files.

Reviewer #3: **Yes: **Joseph Joachim Joseph

---

## [Editor Report · Acceptance letter]

17 Apr 2023

PONE-D-22-07674R2 

Service Availability and Readiness of Malaria Surveillance Information Systems Implementation at Primary Health Centers in Indonesia 

Dear Dr. Veruswati:

I'm pleased to inform you that your manuscript has been deemed suitable for publication in PLOS ONE. Congratulations! Your manuscript is now with our production department. 

Kind regards, 

on behalf of

Dr. Luzia H Carvalho 

Academic Editor

PLOS ONE